# Genome-Wide Identification of Barley ABC Genes and Their Expression in Response to Abiotic Stress Treatment

**DOI:** 10.3390/plants9101281

**Published:** 2020-09-28

**Authors:** Ziling Zhang, Tao Tong, Yunxia Fang, Junjun Zheng, Xian Zhang, Chunyu Niu, Jia Li, Xiaoqin Zhang, Dawei Xue

**Affiliations:** College of Life and Environmental Sciences, Hangzhou Normal University, Hangzhou 311121, China; zhangziling@stu.hznu.edu.cn (Z.Z.); tongtao@stu.hznu.edu.cn (T.T.); yxfang12@163.com (Y.F.); zhengjunjun0415@163.com (J.Z.); zhangxian@hznu.edu.cn (X.Z.); niuchunyu@stu.hznu.edu.cn (C.N.); lijia@stu.hznu.edu.cn (J.L.)

**Keywords:** barley, ABC gene family, gene expression, abiotic stress

## Abstract

Adenosine triphosphate-binding cassette transporters (ABC transporters) participate in various plant growth and abiotic stress responses. In the present study, 131 ABC genes in barley were systematically identified using bioinformatics. Based on the classification method of the family in rice, these members were classified into eight subfamilies (ABCA–ABCG, ABCI). The conserved domain, amino acid composition, physicochemical properties, chromosome distribution, and tissue expression of these genes were predicted and analyzed. The results showed that the characteristic motifs of the barley ABC genes were highly conserved and there were great diversities in the homology of the transmembrane domain, the number of exons, amino acid length, and the molecular weight, whereas the span of the isoelectric point was small. Tissue expression profile analysis suggested that ABC genes possess non-tissue specificity. Ultimately, 15 differentially expressed genes exhibited diverse expression responses to stress treatments including drought, cadmium, and salt stress, indicating that the ABCB and ABCG subfamilies function in the response to abiotic stress in barley.

## 1. Introduction

Named after the binding frame of adenosine triphosphate (ATP), ATP-binding cassette transporters (ABC transporters) are widely found in eukaryotes and prokaryotes [1]. A previous study found that ABC transporters, as one of the most widely functional protein superfamilies, are involved in plant physiological processes [2], such as plant hormone transport, nutrient uptake by organisms, stomatal regulation [3], environmental stress responses, and the interaction between plants and microorganisms [4]. Plant ABC transporters possess nucleotide-binding domains (NBDs) and transmembrane domains (TMDs), and the NBD is a hydrophilic domain with several highly conserved motifs, characterized by the Walker A and Walker B sequences, the ABC signature motif (also known as Walker C) [5], and the H loop, and the Q loop [6]. In contrast to the NBD domain, ABC proteins contain low homologous hydrophobic transmembrane domains (TMDs), and they typically consist of at least six transmembrane α-helices. The NBD domain provides energy through combining and hydrolyzing ATP, whereas the TMD domain is able to select substrates for transportation across membranes through the energy channel provided by the former [7]. TMDs function as selectors for substrates to translocate membrane proteins through the NBD energy channel. A typical ABC full-size transporter have a core unit of two pore-forming TMDs and two cytosolic NBDs [2]. Half-size transporters, composed of only one TMD and one NBD, are thought to form homodimerize or heterodimerize that act as functional pump [8]. In many identified ABC transporters in bacteria, some NBDs and TMDs are present on different polypeptides, called 1/4 molecular transporters [9]. Meanwhile, a few ABC transporters are not directly involved in transport and have been found to participate in other cellular processes such as DNA repair and the transcription and regulation of gene expression [10,11,12,13,14]. In conclusion, although the amino acid sequence of ABC transporters is homologous, the function of ABC transporters is diverse owing to their different structures.

As a consequence of the rapid development of whole-genome sequencing, the ABC gene family had been identified in an increasing number of plants, including 131 in *Arabidopsis thaliana* [15], 125 in rice, 314 in rape [16], and 130 in maize [17]. A large number of plant ABC transporters are plant secondary metabolites that have evolved in response to a particular living environment. Owing to the wide range and large number of ABC transporters, several methods have been proposed to ABC protein nomenclature [18]. According to the homologous relationships, phylogenetic relationships, and domain organization, the new nomenclature system of HUGO system (Human Genome Organization) categorizes ABC transporters into eight subfamilies, ABCA to ABCH subfamily. However, ABCH has not been characterized in plants [19,20]. Afterwards, only ABCI subfamily containing “prokaryotic”-type ABCs has been identified in plants [18]. In total, eight subfamilies (ABCA–ABCG and ABCI) have been identified in plant genomes [21].

It is well universal that abiotic stresses, such as temperature, drought, salt, and heavy metals, seriously restrict plant growth and affect the yield and quality of crops [22]. In severe cases, abiotic stress will directly result in plant death. A growing number of studies have demonstrated that ABC transporters play a pivotal role in crop yield [23], quality formation [24], and the resistance response [25]. Given the importance of ABC transporters in plant life activities, increasingly, plant ABC transporters have been identified, cloned, and functionally analyzed [26]. To date, the ABC gene family has not been identified and analyzed in barley, and its association with abiotic stress has not been explored. Barley (*Hordeum vulgare* L.) is one of the oldest cultivated crops in the world. It is widely adaptable and exhibits strong drought, cold, and salt tolerance characteristics [27]. In the research, we used bioinformatics method to conduct a whole-genome study of the ABC gene family in barley and identified 131 ABC transporters. We performed sequence characteristics, physicochemical properties, gene phylogeny, and expression profile analysis of ABC proteins at the genomic level in barley. We also investigated the expression patterns of barley under cadmium (Cd), drought, and salt stresses by quantitative real-time (qRT)-PCR. Our findings improve understanding of the function of the ABC gene family and will facilitate further studies on detailed molecular and biological functions in barley.

## 2. Materials and Methods

### 2.1. Identification of ABC Gene Family Members in Barley

The ABCs domain-containing protein and genome sequence were retrieved from Pfam database (http://pfam.xfam.org/) [28]. The identified ABC sequences of barley and rice were confirmed for the presence of PFAM domain PF00005 (ABC transport domain) and PF00664 (ABC transmembrane domain) in barley and rice using HMMER program. In order to ensure accuracy analysis, we uploaded conserved sequences into NCBI-CDD [29] and SMART database [30] (http://smart.embl-heidelberg.de/) for protein prediction and unannotated sequences were removed. At the same time, the final protein-coding sequences were verified by searching NCBI non-redundant protein sequence database with BLASTP. Protein features including molecular weight and isoelectric point (pI) of the HvABC proteins were predicted and analyzed by using tools from ExPAsy website (https://web.expasy.org/protparam/) [31,32].

### 2.2. Multiple Sequence Alignment and Phylogenetic Analysis of the ABC Gene Family in Barley

For the sake of understanding the phylogenetic relationship of ABC proteins between barley and rice, the phylogenetic tree was constructed using all the identified ABC amino acid sequences of barley and rice. Multiple alignments of sequences were conducted using MUSCLE [33] of the EMBL-EBI [34] software with the default options. Then, MUSCLE website was utilized to construct the phylogenetic tree by the neighbor-joining (NJ) method with a bootstrap test of 1000-fold (https://www.ebi.ac.uk/Tools/msa/muscle/) [35]. The results were displayed using iTOL visualization.

### 2.3. Analysis of ABC Gene Structure and Chromosome Location in Barley

The information of barley ABC gene family, including intron, exon, physical location on chromosome and gene annotation file information, was retrieved in the Ensemble Plants database.

The exon–intron organizations of all the *HvABC* genes were exhibited using the online program Gene Structure Display Server (http://gsds.cbi.pku.edu.cn/) [36]. The MEME online program for protein sequence analysis was used to identify conserved motifs of ABC proteins (http://meme-suite.org/tools/meme) [37]. The MG2C was used to draw the location images of HvABCs on chromosomes. The protein sequence was used to predict subcellular localization of barley ABC gene family using WoLF PSORT (https://wolfpsort.hgc.jp/) [38].

### 2.4. Construction of ABC Gene Expression Profiles in Barley

To create the expression profile of *HvABC* genes among different organs and development stages, the RNA-seq data from various tissues in barley were retrieved from IPK (https://webblast.ipk-gatersleben.de/barley_ibsc/index.php). The dataset, 14 stages, included the, root (ROO1, ROO2), leaves (LEA, SEN), inflorescences (INF2, LOD, PAL, LEM, RAC), grain (CAR5, CAR15), etiolated seeding (ETI), tillers (NOD), and epidermal strips (EPI). The transcript abundance of *HvABC* genes was calculated as fragments per kilobase of exon model per million mapped reads (FPKM) and log2 (FPKM+ 1) values of invertase genes in these tissues were used to depict heatmaps. The cluster results were shown using the Multiple Experiment Viewer (MeV) (J. Craig Venter Institute, La Jolla, CA, USA).

### 2.5. Stress Treatment, Total RNA Extraction, and qRT-PCR Analysis

The barley cultivar, Morex, was selected for stress treatments. Seedlings were grown on nutrient solution [39] in growth chambers at 26 °C under a 14/10 h light/dark photoperiod and photosynthetically activated radiation at 18,000 lx. Two-leaf-stage plants were treated with different abiotic stress. For drought, salt, and cadmium treatments, the seedlings were treated with 20% PEG6000, 200 mmol·L ^−1^ NaCl and 50 μmol·L ^−1^ CdCl_2_ nutrient solution for 24 h, respectively. All these leaf samples were snap-frozen in liquid nitrogen and the total RNAs were isolated from young leaves using an RNA kit (AxyPrep, USA). Then, the RNA was reverse-transcribed using the HifairTM II 1stStrand cDNA Synthesis Kit following the manufacturer’s instructions. RNA extraction and cDNA synthesis from all samples were stored at −80 °C for RNA extraction.

Based on NCBI, ABC transporters related to abiotic stress, such as rice, wheat and maize, were retrieved. Then, the phylogenetic tree with barley ABC transporter (the method is the same as above) was constructed, combined with the expression information of barley ABC gene, expression site, and gene intron, 15 *HvABC* genes were selected to conduct qRT-PCR. cDNA obtained was used for quantitative RT-PCR using SYBR Green Master Mix and a BioRad CFX96 real-time system. The qRT-PCR experiments were performed with three biological and technical replicates. The relative expression levels were calculated using the formula 2^−∆∆CT^ [40]. The result was analyzed using SigmaPlot v10.0. Primers for qRT-PCR were designed using Primer Premier v5.0., using the website of Ensemble Plants to verify the specificity of primers. The barley actin gene *HvActin* (*HORVU1Hr1G002840*) was used as an internal control. The primers sequence used are listed in Table 1.

## 3. Results

### 3.1. Identification and Physicochemical Properties of the ABC Gene Family in Barley

Through multiple bioinformatics analyses, a total of 131 ABC transporter genes were identified in barley (Appendix A). The barley *ABC* genes were classified according to their sequence similarity with rice *ABC* genes and were further named *HvABCA*–*HvABCG*, *HvABCI*. Among the 131 HvABC proteins, all of the proteins contained one or more NBDs domains based on the domain composition analysis of the ABC proteins.

Comprehensive information on the HvABCs, including the domain structure, predicted protein length, exon number, molecular weight (MW), isoelectric point (PI), and subcellular localization, is provided in Appendix A. The amino acid numbers scoped from 171 aa (HvABCG1) to 1628 aa (HvABCC9), and the corresponding molecular weight changed from 19391.19 to 182783.15 kD. The protein lengths of ABCA, ABCE, ABCF, and ABCI had few differences, while the protein lengths of ABCB, ABCC, and ABCG varied greatly. In contrast, the variation in PI was small, with the majority constituting basic proteins. Based on the subcellular localization prediction of 131 barley *ABC* genes, 99 *ABC* genes were detected to be localized on the plasma membrane, which confirmed that most of the ABC transporters are bound to the plasma membrane and are responsible for the efflux of intracellular substances, while only a few are present on the vacuoles, chloroplasts, and mitochondria in these organelles. ABC transporters mainly regulate the division of exogenic substances into the organelles, which also reflect the endosymbiotic origin of the plastids and mitochondria [41]. Previous studies on the ABCC subfamily have shown that ABCCs are involved in cellular detoxification, which may contribute to the complexation of toxins and heavy metal ions with glutathione or organic acids for storage in the vacuoles or transportation out of cells [42,43]. All 22 genes of the ABCC subfamily in barley reside in the plasma membrane.

### 3.2. Phylogenetic Analysis of the ABC Gene Family in Barley

Previous studies have consistently demonstrated that the ABC gene family underwent species-specific amplification after the divergence between dicotyledonous and monocotyledonous plants [44]. Exploring the relationship between HvABC proteins and OsABC proteins could help classify and assess the potential functions of HvABCs. For further classification, ABC sequences from two different plant species, including 131 HvABC proteins and 100 OsABC proteins, were subjected to phylogenetic analysis. Based on the phylogenetic relationships with OsABCs, the HvABCs were divided into eight subfamilies (ABCA–ABCG, ABCI) (Figure 1). In the light of phylogenetic tree, except ABCI were dispersed, the HvABC proteins of each subfamily were clustered together. Among those subfamilies, the ABCB and ABCG subfamily had the greatest number of members with 32 genes and 49 genes, accounting for 21.97% and 37.12%, respectively. Only three members were identified in ABCE (Appendix A). Further investigation revealed that the HvABCE and HvABCF proteins contained two NBDs but, as expected, no TMD. Although their NBDs domains share sequence homology with other members of the ABC gene family in barley, they are not transposons in the conventional sense and have no obvious transport function. Analysis of the phylogenetic tree terminal branches indicated that there were 67 pairs of orthologous proteins between species, among which the ABCG and ABCB subfamily were the greatest with 23 pairs and 18 pairs, respectively, indicating that the ABC gene family retained very high homology in the evolution of barley and rice. In addition, there were 20 pairs of paralogs, among which the ABCA, ABCB, ABCC, and ABCG subfamilies of barley contained one, four, three, and five pairs, respectively, and there were seven pairs of paralogs in rice; that is, the ABCB, ABCC, and ABCG subfamilies contained two, one, and four pairs, respectively. The above findings can infer that the members of the ABC gene family of barley may have evolved independently and expanded in a species-specific way.

### 3.3. Chromosome Mapping of the ABC Gene Family in Barley

A total of 131 *HvABCs* were mapped on the seven chromosomes, and the remaining three genes were distributed on unanchored scaffolds (Figure 2). Chromosome mapping revealed that the *HvABC* genes were mostly concentrated on or near the end of the chromosomes where they exhibited a high variation in their distribution. In addition, 31 genes, the maximum number, were located on chromosome 3H. On the contrary, chromosome 6H contained only nine genes. As shown in Figure 2, eight *HvABCs* clusters (*HvABCA4/HvABCA7, HvABCB11/HvABCB16/HvABCB22, HvABCB4/HvABCB27, HvABCG29/HvABCB23, HvABCC22/HvABCC2, HvABCC6/HvABCC8, HvABCC16/HvABCC15/HvABCI1,* and *HvABCC13/HvABCG1*) containing 18 genes were identified on chromosomes 1H, 3H, 4H, and 7H. Tandem duplication was an important recent gene duplication pattern in the expansion of HvABCs gene family. These may be caused by tandem repeat genes. Therefore, these gene clusters may be tandem repeat arrays.

### 3.4. Analysis of Exon-Intron Structure and Conserved Domain of the ABC Gene Family in Barley

To further explore the conservation and diversity of protein structures and compositions of HvABCs, the corresponding analysis is carried out by MEME and GSDS website. The result (Appendix A) showed that most of the HvABC proteins (122 of 131, 93.1%) presented multiple introns, while nine *HvABC* genes (*HvABCG14, HvABCG15, HvABCG12, HvABCG26, HvABCG16, HvABCG28, HvABCG17, HvABCG3, HvABCG1, HvABCI7,* and *HvABCI2*) had no introns. These nine genes may have originated from the transposable events of reverse transcription. The number and size of the introns varied in the remaining 122 genes, indicating significant differences in gene structure. It can also be speculated that intron deletion and insertion events occurred in the evolutionary process, which may be caused by the differentiation of ABC gene family members after replication. The exons of all identified *HvABC* genes ranged from 1 to 30, among which the number of *ABC* genes with 10 exons was the highest, accounting for 9.92%.

MEME motif analysis identified seven conserved motifs in the HvABC proteins. Combining the domain characteristics of each subfamily, we analyzed the conservative motifs (Appendix A). The number of conserved motifs in each HvABC protein varies from one to five. The results (Figure 3) indicate that all seven highly conserved motifs belong to the NBD domain, which also suggests that the NBD sequence identity was higher than that of the TMDs. Among them, motif 2 is the Walker A in the nucleotide binding domains, and motif 6 is the Walker B in the nucleotide binding domains, and the motif between Walker A and Walker B is the ABC characteristic motif; that is, motif 1, motif 3, motif 4, motif 5, and motif 7. However, the [LIVMFY] subunit in motif 1 contains other residues (Table 2), and motif 6 in Walker B is interrupted by a hydrophilic residue. This phenomenon is currently only observed in plant ABC transporters [45]. The result also shows that several motifs are widely distributed in the HvABC proteins, such as motifs 1 and 2. In contrast, other motifs are specific to only one or two subfamilies. For instance, only ABCB and ABCC subfamilies contain motif 3, and motif 7 exists only in ABCC and ABCI. ABCG and ABCF contain the specific motif 4, and these motifs are probably required for specific protein functions. The functional differentiation in HvABCs during the evolutionary process may be due to the diversity of motif components in the different subfamilies.

### 3.5. Tissue-Specific Expression of ABC Genes in Barley

Tissue-specific expression patterns of genes can help elucidate their function in plant species and predict their role in growth and development. To further elucidate the expression profiles of *HvABCs* in different tissues and developmental stages, we used expression data of different tissues from the IPK website as a resource. A heatmap displaying the expression data of *HvABCs* in different periods as well as in the tissues and organs was generated, and the *HvABCs* were clustered by their expression patterns. Tissue specific expression profiles showed that the *HvABC* genes could be categorized into eight types (Figure 4), which indicated that the expression of *ABC* genes had undergone significant differentiation. Furthermore, the expression data clustering results did not clearly correspond to the subfamilies differentiated by the phylogenetic analysis, indicating that sequence similarity does not completely determine expression pattern and function similarity.

In this study, a total of 127 *HvABCs* were determined as being expressed in at least one tissue (the FPKM values of *HvABCB1*, *HvABCC3*, *HvABCC4*, and *HvABCG9* in the 15 tissues were all 0), and 92 *HvABCs* were expressed in all tissues. The tissue-specific expression profile showed that some HvABC gene family members did not have tissue specificity and highlighted an essential role in almost all growth and developmental stages. As the Figure 4 shows, the majority of *HvABCs* presented different expression patterns, whereas a few exhibited similar expression patterns. Some ABCC subfamily genes were ubiquitously and highly expressed in all the tissues, especially the root, such as *HvABCC1*, *HvABCC7*, and *HvABCC18.* Some *HvABCs* also exhibited tissue-specific expression; for instance, *HvABCG10* only specifically expressed in the leaf tissues, *HvABCB11* and *HvABCB16* specifically expressed in the developing grain, and *HvABCG12*, *HvABCG14*, and *HvABCG29* were high during tiller development, implying that these genes may play specific roles in the relevant tissues.

### 3.6. Expression Analysis of ABC Genes in Barley in Response to Abiotic Stress

Research has found that plant ABC transporters play a major role in auxin, heavy metal transport, and abiotic stress, especially the ABCG subfamily and ABCB subfamily [46]. In this study, a total of 15 *HvABC* genes were identified using bioinformatics methods, including one *HvABCA*, two *HvABCBs*, two *HvABCCs*, two *HvABCFs*, and eight *HvABCGs*. The expression responses of selected *HvABC* genes under NaCl, PEG, and Cd treatment conditions were examined using qRT-PCR in our study.

A range of expression levels were observed in the selected *HvABCs* at 24 h after exposure to Cd, drought, and salt stress. The analyses revealed that different members within each HvABC subfamily responded differently to the same set of abiotic stresses. Compared with the control levels, the results showed that the expression levels of 13 of the 15 identified *HvABC* genes increased in response to Cd stress. The most pronounced increases were observed in *HvABCA3* and *HvABCG29*, whereas *HvABCG38* and *HvABCG21* were dramatically repressed by Cd stress (Figure 5A). Under salt stress conditions, most *HvABC* genes exhibited the opposite pattern to Cd stress, with *HvABCG48*, *HvABCF4*, *HvABCB24,* and *HvABCG25* being repressed (Figure 5B). Expression analysis following PEG treatment indicated that *ABC* genes were upregulated by 1.0 times, including *HvABCG45, HvABCG48, HvABCA3, HvABCF5, HvABCF4, HvABCC16, HvABCC11,* and *HvABCB13.* Under different abiotic stresses, *HvABC* genes showed different response patterns and different response degrees. Among the 15 *HvABC* genes, *HvABCG38* was repressed after all the stress treatments, whereas the expression of other genes, including *HvABCG45*, *HvABCA3*, *HvABCF5, HvABCC16, HvABCC11, HvABCB13, HvABCG27, HvABCG29,* and *HvABCG18,* was increased by the stress treatments (Figure 5C). Conversely, the expressions of *HvABCG4*, *HvABCG25*, *HvABCG21*, *HvABCF4,* and *HvABCB24* were enhanced or induced by the stress treatments.

## 4. Discussion

The ATP-binding cassette (ABC) transporters belong to a large superfamily of proteins, which are ubiquitous and important in all kind of life events for all life organism [9,47]. In plants, ABC transporters participate in the transport of exogenous substances and secondary metabolites and in the abiotic stress response, as well as in many other important physiological and developmental processes [6,21].

Because of repeated genome replication, the number of ABC proteins in plants is much higher than in animals. Given the important regulatory role of ABC transporters in plant growth and development, bioinformatics analysis of ABC family genes has been conducted in *Arabidopsis* [15], rice [48], maize [17], *Lotus corniculatus* [49], grape [50], *Brassica napus* [16], and other important plants. Thus far, most research into the functions of ABC transporters has been concentrated on *Arabidopsis*. In this study, 131 HvABC proteins were identified genome-wide in barley (Appendix A), which is similar to the number of genes identified in rice and *Arabidopsis* [15,48]. Additionally, phylogenetic analysis divided the protein sequences of barley and rice into eight subfamilies, including ABCA–ABCG and ABCI (Figure 1). In each subfamily, the *ABC* orthologous genes reflect the sequence similarity of barley and rice, and relatively, the emergence of paralogous genes indicates differentiation It is presumed that the barley genome experienced whole-genome replication events before the separation of gramineous species. All *HvABC* genes have at least one NBD domain, which indicates that NBD is a unique domain of ABC transporters. The HvABC transporter domain has various organizational forms that can be split into full-size members, half-sized transporters, and 1/4 molecular transporters [8,9]. *HvABC* include multi-domains, whereas few genes generally contain only one NBD domain. This phenomenon has been confirmed in *SLABCB27* of the tomato ABC gene family [51]. Conserved sequence analysis also reflects the uniqueness of the ABC domain and the high conservation of NBDs. On the contrary, the sequences and structures of TMDs differ, which reflects the chemical diversity of the substrates transported by ABC transporters. The members of the HvABC gene family have large functional differences, with large span amino acid, as well as large differences in isoelectric point and pH value (Appendix A). In short, the structural differences also reflect the different functions.

Gene expression pattern is an important starting point for further evolution and function research. Thus, we conducted expression analysis for all HvABCs in 15 tissues and organs. Except for a few genes, most HvABCs did not exhibit clear tissue specificity (Figure 4), indicating that HvABCs play a role in all aspects of plant growth. Studies have shown that *Arabidopsis AtABCB19* participates in the regulation of the separation of post-embryonic organs and cytoplasmic flow in the inflorescence axis of *Arabidopsis* [52,53], while *AtABCB1* plays a major role in another development, and *AtABCB19* plays a synergistic role [54]. The *AtABCG11* protein is distributed in the stems, leaves, and floral organs of plants and is involved in the transportation of paraffin wax and grease substances on the surface of plants.

Previous studies have found that the ABCB subfamily may be closely associated with abiotic stress in plants [17,55,56]. For example, *AtABCB25* is related to heavy metal resistance in *Arabidopsis,* and the overexpression of *AtABCB25* can improve the resistance of *Arabidopsis* to Cd and lead [57]. *OsABCB23* and *OsABCB24* are induced by drought stress, while *OsABCB6*, *OsABCB9,* and *OsABCB8* are induced by salt stress [48,58]. *OsABCB27,* located on the vacuolar membrane, participates in the response of rice to aluminum stress [59]. Under drought stress, the expression of the *ZmABCB7* and *ZmABCB8* genes increases significantly, while the expression of *ZmABCB18* is significantly inhibited by salt stress [17]. In this study, it was found that the expression of *HvABCB13* was significantly increased under Cd, drought, and salt stress, while *HvABCB24* was inhibited by salt stress and drought, but could respond significantly to Cd stress. ABCG transporters excrete wax and keratinous monomers, thus reducing water loss [60]. *AtABCG12* participates in the transmembrane transport of cuticular wax in the stem epidermis, *AtABCG32* is involved in the formation of the cell wall cuticle, and the *AtABCG32* transporter substrate is a keratinous monomer. When plants suffer from drought stress, *AtABCG40* increases abscisic acid production to timeously close the stomata [61]. At the same time, as the largest subfamily, ABCG also plays an important role in abiotic stress. *SpTUR2* in duckweed is expressed upon exposure to salicylic acid, cold, and high-salt environments [62]. The overexpression of the *AtABCG36* gene can improve the resistance of *Arabidopsis* to salt and drought [63]. The ABCG transporter *OsABCG9* in rice roots responds significantly to PEG, zinc, and Cd stress, and its expression is closely related to redox mechanism and heavy metal stress [64]. Our study shows that the HvABCG subfamily responded to salt, drought, and Cd stress to different degrees. *HvABCG45*, *HvABCG27*, *HvABCG29,* and *HvABCG18* were concurrently involved in three types of stress induction. In addition, *HvABCG38* and *HvABCG21* could significantly respond to salt stress, and their expression levels decreased significantly after drought and Cd treatments. *HvABCG48* and *HvABCG25* were significantly upregulated under drought and Cd stress but were inhibited by salt stress (Figure 5). This also confirms that the HvABCB and HvABCG subfamily are involved in the abiotic stress response, and most HvABC genes are regulated by at least one abiotic stress factor.

A genome-wide analysis of ABC gene family in barley was carried out in the present study. The phylogenetic relationships, gene structures, chromosome locations, and expression profiles of *HvABCs* were studied in detail. Furthermore, 15 genes responding to abiotic stress were preliminarily screened, and the expression levels of the genes following stress treatment were analyzed. Taken together, our study revealed the functional diversity of HvABCs proteins and provided candidate *ABC* genes for future breeding to various stresses.

## Figures and Tables

**Figure 1 plants-09-01281-f001:**
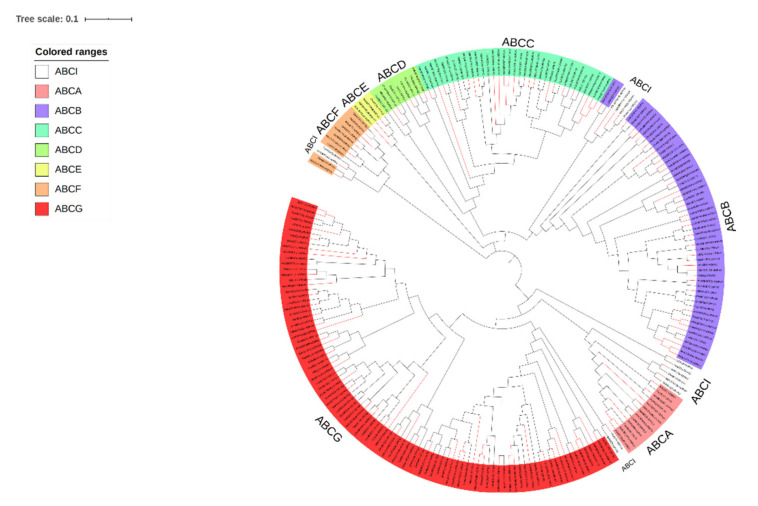
Unrooted Neighbor-Joining tree constructed with ABC proteins of *Hordeum vulgare* L. (HORVV) and *Oryza sativa* L. (ORYSJ). The domains clustered into eight subgroups (ABCA-ABCG, ABCI). Different colored shadings indicated eight ABC transporter subfamilies. The red branch is barley, and the black branch is rice.

**Figure 2 plants-09-01281-f002:**
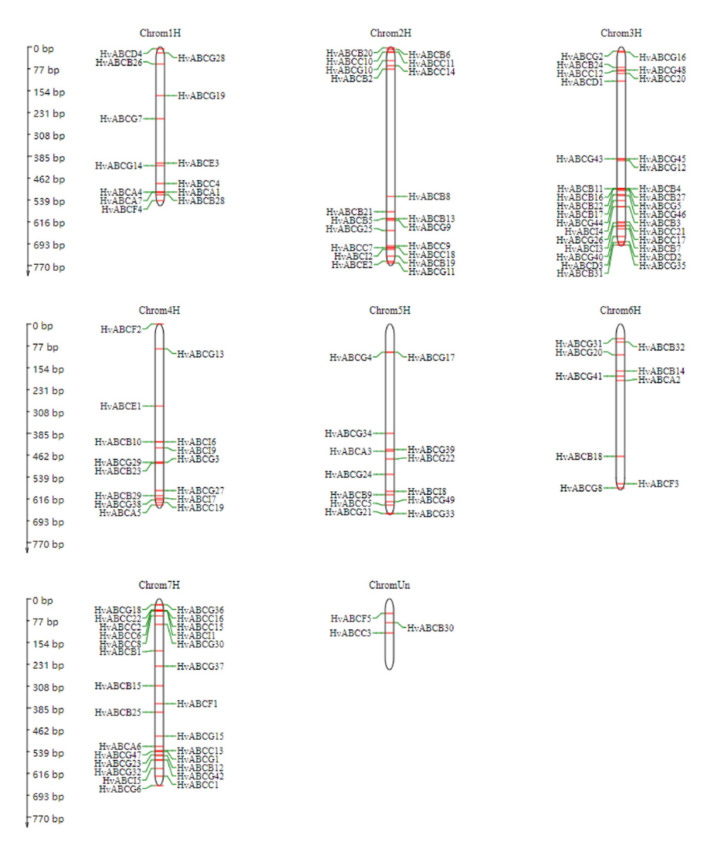
Mapping of the *HvABC* genes on *Hordeum vulgare* L. chromosomes. The chromosome number is indicated at the top of each chromosome. Three putative *HvABC* genes could not be localized on a specific chromosome.

**Figure 3 plants-09-01281-f003:**
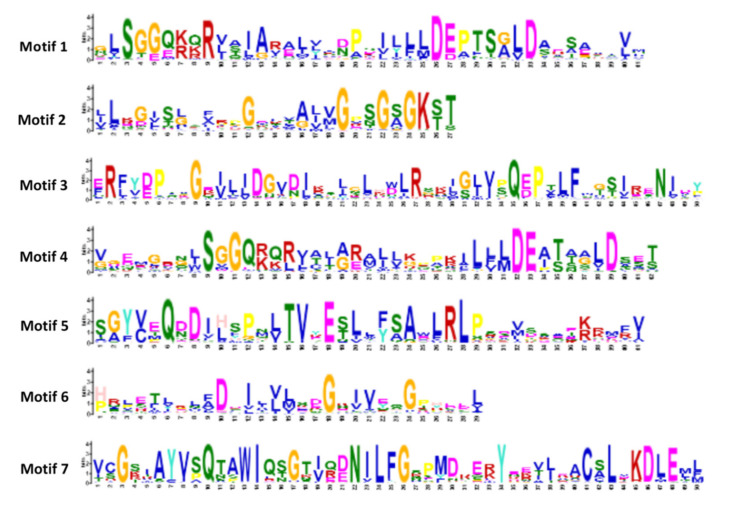
Conserve amino acid in seven motifs of ABC gene family in barley. Motif analysis and the sequence logos was performed using MEME website.

**Figure 4 plants-09-01281-f004:**
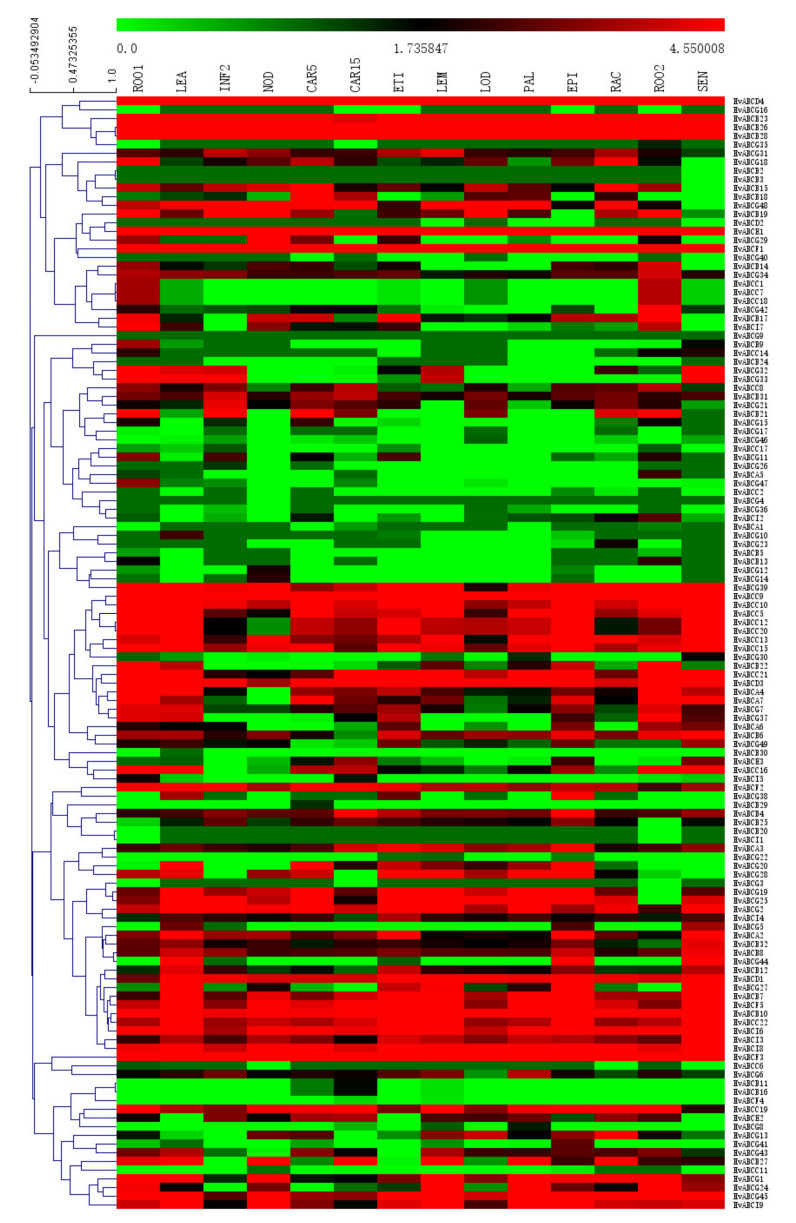
Heatmap showing the expression pattern of *HvABC* genes in developmental stages and tissues, including ROO1 (Roots from seedings), ROO2 (Roots), LEA (Shoots from seedings), ETI (Etiolated seeding, dark cond), INF2 (Developing inflorescences), PAL (Dissected inflorescences), LEM/LOD/RAC (inflorescences, lemma/lodicule/rachis), NOD (Developing tillers), CAR5/CAR15 (Developing grain, 5 DAP/15 DAP), EPI (Epidermal strips), SEN (Senescing leaves). The combined phylogenetic trees of *HvABCs* genes on the left panel. The scale bar at the top represents relative expression value. Red denotes high expression levels, and green denotes low expression levels.

**Figure 5 plants-09-01281-f005:**
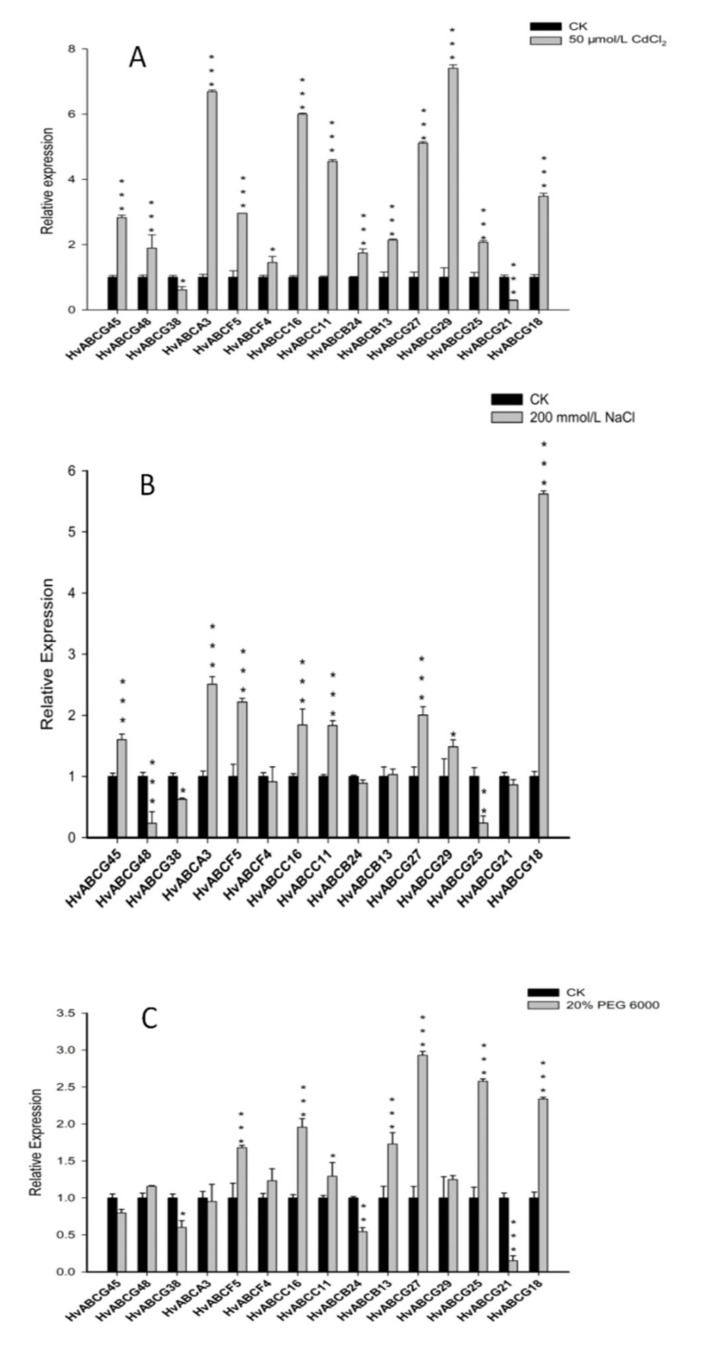
qRT-PCR analysis of 15 *HvABC* genes in response to (**A**) 50 μmol·L^−1^ CdCl_2_ (**B**) 200 mmol·L^−1^ NaCl (**C**) 20% PEG6000. columns in black represent CK, columns in gray represent abiotic stress. ANOVA and LSD was used to test significance. Asterisks indicate the corresponding gene significantly up- or down-regulated compared with the untreated control (* *p* < 0.05, ** *p* < 0.01, *** *p* < 0.001) Data are the means of three replicates with standard errors represented by bars.

**Table 1 plants-09-01281-t001:** Primer sequence used for qRT-PCR amplification of ABC gene family in barley.

Gene Name	Primer	Forward Primer Sequence (5′-3′)	Reverse Primer Sequence (5′-3′)
*HvABCG45*	ABC1	GGCGGAACTGCTGTCATCT	AGTCGGCAACACCCTTTCT
*HvABCG48*	ABC2	CAGCCTGGGTTCGTTTGAG	TCGGAGTGATCGCCGTTGT
*HvABCG38*	ABC3	GGTTTGGATGCTCGTGCTG	TGATTTGACCGCCTCTTTT
*HvABCA3*	ABC4	TGGGCTCATTCCACCTACA	CCGTCAATGTTTCCCAGAG
*HvABCF5*	ABC5	TGGCTGGAAGAAACACTGAA	TCGGGTCTGCACATACTGGTC
*HvABCF4*	ABC6	CACATGCAGAACAAGACCCTC	GCTTCGCAGATCCATGACC
*HvABCC16*	ABC7	GCCATTCGGCGACCATACA	CACGAGCAAGCTGAACACG
*HvABCC11*	ABC8	GTCCTTGACGCTGATACTGG	TAGCACTGGTGCCTCCTCC
*HvABCB24*	ABC9	TGATACTGGGATTTGGTTAGG	CGAATGGCACTGAGAATGAG
*HvABCB13*	ABC10	CGTTCAACTCGGAGGACAAGA	CCATGCAGCGTACCACAGG
*HvABCG27*	ABC11	GAGGGAGGCAGCGTCAAGCA	GCAGGATGGCGAACTGGTTG
*HvABCG29*	ABC12	AGGGCTTCCCGTTGTAGGTG	TCGCATCCGTCATCACCATG
*HvABCG25*	ABC13	GTTCTGGATCGAGATGGGTGT	GAAGATGGTCGCCAGGATGA
*HvABCG21*	ABC14	ATACCGGCATACTGGCTGTGG	CCAGCACTCGCTCCTTCACC
*HvABCG18*	ABC15	TGCTCACCGCCAACTCATTC	TCCTTGCTCGCCACGAAGT
*HvActin*	Actin	TGGATCGGAGGGTCCATCCT	GCACTTCCTGTGGACGATCGCTG

**Table 2 plants-09-01281-t002:** Seven conservative motif protein sequences given by the MEME online tool.

Motif	Protein Sequence
Motif1	GLSGGQKQRVAIARALLABPSILLLDEPTSGLDAESAAIVM
Motif2	LLKGISLSFRPGELVALVGPSGSGKST
Motif3	ERFYDPTAGEILJDGVDIKSJGLHWLRSKJGJVPQEPTLFMGSIRENIDY
Motif4	VGEMGRNLSGGQRQRVALARAJLKPPKILLLDEATAALDSET
Motif5	SGYVEQBDIHSPNLTVYESLLFSAWLRLPSDVSSAEKRMFV
Motif6	HRLETLRLFDDILVLSDGKIVEQGPHEEL
Motif7	VCGSIAYVSQTAWIQSGTIQDNILFGSPMDRERYEEVJEACSLVKDLEML

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
