# Peer review of "Genome-Wide Identification of Barley ABC Genes and Their Expression in Response to Abiotic Stress Treatment"

_plants, 2020, doi:10.3390/plants9101281_

Round 1

Reviewer 1 Report

The authors try to understand the function of ABC gene family in barley. The investigate using a bioinformatic methodology like multiple sequence alignment, motif/domain identification, phylogeny analysis etc.

Comments:

  1. A very similar article can be found in https://doi.org/10.1186/s12870-019-1681-6 . I couldn't find any extra steps from the authors to study anything more in their own way. The only difference between both the studies is the species and gene. I would like authors to contribute their own structure and aim to the work apart from the article mentioned above. 
  2. Why do the authors use clustalw and muscle together? Which alignment tool do they rely on at the end?
  3. Since it is a novel identification why the MSA done using NJ and not using ML?
  4. 2.5 section were not correctly mentioned about the stress treatment. How they did the treatment, how long was the treatment done, the stage of the sample before and after collecting the sample?
  5. Is there any reason to consider barley and rice together for the identification of the ABC gene family?
  6. The heading for introduction should be changed to 1. instead of .11.
  7. I missed a detailed explanation or in-depth study about 2.5 which is one of the major scientific parts of the manuscript.
  8. The study is found to be an overview of much smaller analysis but missing the in-depth descriptive and detailed study. I believe authors can give more insight into the study and make the aim more scientific.

Author Response

1.A very similar article can be found in https://doi.org/10.1186/s12870-019-1681-6 . I couldn't find any extra steps from the authors to study anything more in their own way. The only difference between both the studies is the species and gene. I would like authors to contribute their own structure and aim to the work apart from the article mentioned above.

Response: Thanks. We cited this paper in the revised manuscript.

2.Why do the authors use clustalw and muscle together? Which alignment tool do they rely on at the end?

Response: Thank you. In this experiment, Pairwise comparison analysis was performed by Clustal W. However, due to the large number of HvABC genes, the accuracy and speed of MUSCLE were better than that of Clustal W, so MUSCLE was used to construct the phylogenetic tree. The final alignment tool depends on Clustal W.

  1. Since it is a novel identification why the MSA done using NJ and not using ML?

Response: Thank you. NJ is based on the principle of minimum evolution. The construction of phylogenetic tree is relatively accurate and the calculation speed is fast. There is only one tree, which can analyze more sequences, and the running speed is better than ML. So we chose to use the NJ method to construct the phylogenetic tree.

  1. 2.5 section were not correctly mentioned about the stress treatment. How they did the treatment, how long was the treatment done, the stage of the sample before and after collecting the sample?

Response: Thank you. The seedlings were treated with 20% PEG6000, 200mmol·L -1 NaCl and 50 μmol · L - 1 CdCl2 nutrient solution and the treatment time was 24 hours. The stage before and afte rcollecting the sample is the two-leaf stage. In addition, I have supplemented the corresponding information in the article.

  1. Is there any reason to consider barley and rice together for the identification of the ABC gene family?

Response: Thank you. Studies have demonstrated that the ABC gene family underwent species-specific amplification after the divergence between dicotyledonous and monocotyledonous plants. Both barley and rice are monocotyledons. The research on related genes in rice is thorough and it is related to barley. Thus, building phylogenetic trees from rice and barley together can better understand the function and evolutionary status of HvABC genes.

  1. The heading for introduction should be changed to 1. instead of .11.

Response: Thank you for your suggestion. I have rechecked and changed in accordance with the requirements of the journal.

  1. I missed a detailed explanation or in-depth study about 2.5 which is one of the major scientific parts of the manuscript.

Response: Thank you. We revised.

  1. The study is found to be an overview of much smaller analysis but missing the in-depth descriptive and detailed study. I believe authors can give more insight into the study and make the aim more scientific.

Response: Thanks. We revised.

Reviewer 2 Report

This paper to me is biologically sound, but I have two major comments on the statistical analysis. Figure 4 needs to be rearranged to support the conclusions, and the statistics in Figure 5 need to be redone.

Major Comments:

  1. In Figure 4, could you try manually arranging the genes by their subfamily? Right now, I cannot see that the subfamilies are not clustering together because they are not marked. I think that clustering by subfamily rather than expression will better support your conclusions.
  2. In Figure 5: I am not understanding why ANOVA was used. Also, your ANOVA results do not make sense. You cannot just report p-values, you must do a post-hoc test.

    You are looking at the effect of stress on each gene. In this case, you should use a pairwise t-test on each of the genes separately. So, please redo the statistics by performing a pairwise t-test on each gene, CK vs PEG.

Minor Comments:

  1. The section header for the Introduction should be 1.1 (there is a typo)
  2. Line 121, "performed" should be "treated"
  3. Line 122, "liquid" I think should be "liquid nitrogen"
  4. Line 125, this is confusing, cDNA is usually stored at -20 but RNA should be stored at -80, otherwise it will degrade
  5. Line 126-129, multiple grammar errors in this sentence making it hard to understand
  6. I think Table 1 should be supplemental
  7. It would be nice if the font in Figures 1 and 2 could be just a little bit bigger.
  8. Table 2 and Figure 3 are basically providing the same information. I would keep Figure 3 in the main text and move Table 2 to the supplement.
  9. Figure 4 text needs to be increased, or less genes should be included. I cannot read the gene labels at all.
  10. In Figure 5, the text is distorted.

Author Response

Major Comments:

1.In Figure 4, could you try manually arranging the genes by their subfamily? Right now, I cannot see that the subfamilies are not clustering together because they are not marked. I think that clustering by subfamily rather than expression will better support your conclusions.

Response: Thank you for your suggestion. The classification of subfamily is based on sequence similarity, the expression data clustering standard is inconsistent with the standard for phylogenetic group clustering. Sequence similarity does not completely determine the similarity of expression patterns. The eight subfamilies were not clustered in the same way as evolutionary analysis.

2.In Figure 5: I am not understanding why ANOVA was used. Also, your ANOVA results do not make sense. You cannot just report p-values, you must do a post-hoc test.

You are looking at the effect of stress on each gene. In this case, you should use a pairwise t-test on each of the genes separately. So, please redo the statistics by performing a pairwise t-test on each gene, CK vs PEG.

Response: Thank you. Quantitative data is expressed as mean±SD and analyzed by one-way ANOVA. Multiple comparison between the groups was performed using LSD method. Statistical significance was set at a level of P < 0.05. I have improved the statistical method and redo the Figure 5.

Minor Comments:

The section header for the Introduction should be 1.1 (there is a typo)

Line 121, "performed" should be "treated"

Line 122, "liquid" I think should be "liquid nitrogen"

Line 125, this is confusing, cDNA is usually stored at -20 but RNA should be stored at -80, otherwise it will degrade

Line 126-129, multiple grammar errors in this sentence making it hard to understand

I think Table 1 should be supplemental

It would be nice if the font in Figures 1 and 2 could be just a little bit bigger.

Table 2 and Figure 3 are basically providing the same information. I would keep Figure 3 in the main text and move Table 2 to the supplement.

Figure 4 text needs to be increased, or less genes should be included. I cannot read the gene labels at all.

In Figure 5, the text is distorted.

Response: Thank you for your suggestion. I have made relevant revision to the article.

Round 2

Reviewer 1 Report

The authors made corrections throughout the paper. But I still miss the uniqueness for the paper.

As I mentioned before I could not see that extra step from author’s part to include something new which makes the study unique. Just a citation to the similar article is not what is expected.

I got confused with the answer towards the 2nd question. The authors said they used both the pairwise alignment tool (clustalw and muscle). If they rely on MUSCLE for phylogeny which is the end product of the analysis, how can they say the final alignment they rely on was CLUTALW?

Also, the calculation speed of a tool is the less interested parameter for a research when one is looking for the accuracy of the result. Therefore, the reply for NJ over ML is not acceptable.

Author Response

1.As I mentioned before I could not see that extra step from author’s part to include something new which makes the study unique. Just a citation to the similar article is not what is expected.
Response: Thank you. In this paper, a systematic bioinformatics analysis of barley ABC gene family was carried out, and 15 HvABC genes were selected to analyze their expression under abiotic stress (drought, salt, cadmium), which further proved the role of HvABC genes in abiotic stress, and also provided useful clues for elucidating the mechanism of barley ABC gene family in stress response. Abiotic stress analysis is different from other articles.

2.I got confused with the answer towards the 2nd question. The authors said they used both the pairwise alignment tool (clustalw and muscle). If they rely on MUSCLE for phylogeny which is the end product of the analysis, how can they say the final alignment they rely on was CLUTALW?
Response: Thank you. MUSCLE has been re-used for sequence alignment, which is consistent with ClustalW, and the article has been modified accordingly. The final alignment tool depends on MUSCLE.

3.Also, the calculation speed of a tool is the less interested parameter for a research when one is looking for the accuracy of the result. Therefore, the reply for NJ over ML is not acceptable.

Response: Thank you. The ABC transporters is a large and diverse protein family. In this study, 131 and 100 ABC transporters were identified in barley and rice to construct the phylogenetic tree. NJ method is based on the principle of minimum evolution, with few assumptions and accurate tree construction, and which is suitable for the analysis of large number of sequences. In addition, ML depends on the existence of a clear evolutionary model, but the evolution process of ABC in barley and rice still remain complexity with unclear evolutionary model. Therefore, NJ method is suitable for constructing phylogenetic tree of barley and rice.

Round 3

Reviewer 1 Report

Authors were now clearly replied to my queries and modified the manuscript. Thank you for that.